# Salivary and Serum Liquid Biopsy Biomarkers for HPV-Associated Oral and Oropharyngeal Cancer: A Narrative Review

**DOI:** 10.3390/jcm14217598

**Published:** 2025-10-26

**Authors:** Saman Warnakulasuriya, Shankargouda Patil

**Affiliations:** 1Faculty of Dentistry, Oral & Craniofacial Sciences, King’s College London, London SE1 9RT, UK; s.warne@kcl.ac.uk; 2College of Dental Medicine, Roseman University of Health Sciences, South Jordan, UT 84095, USA

**Keywords:** liquid biopsy, HPV, Oral Squamous Cell Carcinoma, saliva, serum, circulating tumor DNA

## Abstract

**Background**: Human papillomavirus (HPV)-associated oral and oropharyngeal squamous cell carcinomas have risen dramatically in incidence over recent decades. Yet, unlike cervical neoplasia, there is no established screening paradigm for HPV-driven oropharyngeal dysplasia, as precursor lesions are often occult and are not easily accessible for examination. This drives an urgent need for non-invasive biomarkers to enable early detection, risk stratification, and timely intervention. Objective of this review is to highlight advances in liquid biopsy modalities, specifically saliva- and blood-based biomarkers—in the context of HPV-driven oral carcinogenesis—and to evaluate their utility in early cancer detection, prognostic, post-treatment surveillance, and recurrence monitoring. **Methods**: We performed a narrative review of PubMed-indexed studies (2015–2025) focusing on HPV-positive oral and oropharyngeal squamous cell carcinomas. and liquid biopsy analytes. Key sources were high-impact original studies and meta-analyses from 2020–2025 examining circulating tumor DNA (ctDNA), viral nucleic acids, circulating tumor cells (CTCs), extracellular vesicles (EVs), and related biomarkers in saliva and blood. Reported data on assay performance, biases, and validation were reviewed to highlight how oral cancer findings align with trends seen in other solid tumors. **Results**: In reviewing recent studies (2015–2025), we found consistent evidence that saliva best captures locoregional tumor signals while plasma circulating tumor HPV DNA (ctHPV DNA) reflects systemic disease, and that using both matrices improves detection over either alone. Dual-fluid testing will potentially enable earlier identification of molecular residual disease with clinically meaningful lead time before radiographic recurrence, supporting risk-adapted surveillance. Overall, literature favors standardized pre-analytics and combined saliva plus plasma workflows to enhance early detection and follow-up in HPV-positive oral and oropharyngeal squamous cell carcinomas. **Conclusions**: Liquid biopsy approaches offer promising tools for the early, non-invasive detection and real-time monitoring of HPV-associated oral cancers. Realizing their full clinical potential will require robust prospective validation and standardization of pre-analytical protocols. Integrating salivary and blood biomarkers into tailored surveillance programs may further support earlier intervention and improved patient outcomes, while potentially reducing reliance on unnecessary invasive procedures.

## 1. Introduction

Oral and oropharyngeal squamous cell carcinomas (OSCC/OPSCC) have entered an epidemiologic transition in the HPV era. While smoking-related oral cancers have gradually declined in some regions, HPV-driven OPSCC has surged, especially among young adults in high-income countries [1,2]. By 2012, the annual number of HPV-positive oropharyngeal cancers in the U.S. surpassed cervical cancers [1]. This viral oncogenesis pathway, predominantly involving high-risk HPV16, yields tumors with distinct biology and better treatment responsiveness compared to traditional OSCC [1]. Nonetheless, the rising burden of HPV-associated disease presents new challenges: these cancers often arise in cryptic locations (tonsillar crypts, base of tongue) and lack a defined potentially malignant phase amenable to screening. A 2025 meta-analysis of 7289 OSCC cases in Asia reported an overall HPV prevalence of 23.1%, highest in South Asia (27.1%), with HPV-16 (9.1%) and HPV-18 (5.1%) predominating; buccal mucosa (34.0%) and floor of mouth (33.2%) showed the greatest site-specific positivity [3]. In contrast to cervical neoplasia where Pap smears and HPV tests enable early lesion detection, there is no screening test for HPV in the throat and no routine clinical examination can reliably visualize early oropharyngeal malignancy [4]. Consequently, many patients present with lymph node metastases despite small primary tumors.

Liquid biopsy, the molecular analysis of tumor-derived biomarkers in body fluids, has emerged as a promising strategy to fill this early detection gap. Recent prospective evidence supports plasma circulating cell-free DNA (ccfDNA) quantification in Head and Neck Squamous Cell Carcinoma (HNSCC): baseline ccfDNA levels are higher in patients than controls, lower post-treatment levels associated with longer progression free survival (PFS), and fluorometry correlates strongly with Quantitative Polymerase Chain Reaction (qPCR), underscoring clinical utility for diagnosis, monitoring, and prognosis [5]. According to the current European Society for Medical Oncology (ESMO) guidelines on the use of liquid biopsies in oncology, validated ctDNA assays are recommended for routine genotyping in advanced-stage cancers such as lung, breast, and gastrointestinal malignancies [5]. A comparative overview of biomarker performance across these tumor types is presented in Table 1. Using the Parsortix^®^ (Guildford, UK) PC1 system, ≥1 CTCs was detected in 45.3% of metastatic breast cancer patients versus 6.9% of controls [6]. HPV-driven oral carcinogenesis is an ideal candidate for liquid biopsy testing: the oncogenic virus itself provides tumor-specific markers (HPV DNA, RNA, and antibodies), and the affected anatomic sites (oral cavity and oropharynx) are bathed in easily obtainable fluids (saliva) [7]. Tumorigenic HPV infection integrates into host cell genomes or causes expression of viral oncoproteins (E6, E7), which inactivate p53 and Retinoblastoma protein (Rb) and dramatically reshape the host cell’s genome and epigenome [8]. Unlike tobacco-carcinogen-driven OSCC, which accumulates many mutations (e.g., TP53, CDKN2A loss), HPV-positive tumors often have a “viral” mutational landscape consisting fewer mutations overall, with recurrent PIK3CA mutations and unique viral-host DNA junctions from HPV integration [8]. These molecular differences create discrete biomarkers: wild-type TP53 with viral E6 oncoprotein expression, overexpression of p16 (a surrogate of Rb inactivation), and circulating fragments of HPV DNA or HPV–human chimeric sequences that can be detected in saliva or plasma [9]. Moreover, the host mounts an immune response to E6/E7, producing serum antibodies that can serve as indirect markers of HPV oncogenesis [10].

In principle, a multi-tiered liquid biopsy approach could revolutionize oral cancer care. For early detection, a simple oral rinse or saliva test that identifies high-risk HPV infection or tumor-specific mutations could enable “HPV-Pap” screening for the oropharynx, analogous to cervical screening [11]. Already, HPV16 DNA detection in oral rinses has been associated with risk for OPSCC, and intriguingly, HPV16 E6 serology can predict oropharyngeal cancer risk years before diagnosis (e.g., 10-year risk ~20% for HPV16 E6 antibody-positive individuals) [12]. For risk stratification, liquid biopsy may help identify which HPV infections are likely to persist and progress. For example, rising levels of cell-free HPV DNA or oncoprotein expression in saliva might flag clonal progression in an otherwise occult lesion. Similarly, persistently elevated anti-E6/E7 antibody titers post-therapy signal residual disease risk [10]. For post-treatment surveillance, sensitive blood assays for circulating tumor HPV DNA can detect minimal residual disease (MRD) well before it forms radiographically detectable recurrence [13]. This could prompt earlier salvage treatment when tumor burden is low, potentially improving outcomes. Finally, for recurrence monitoring, serial liquid biopsies may offer a non-invasive adjunct to imaging, reducing the reliance on blind biopsies or excessive body/organ scans in follow-up [14].

In this review, we outline the biological rationale for liquid biopsies in HPV-associated oral cancer, emphasizing how tumor biology (viral oncoproteins, DNA shedding mechanisms) intersects with saliva and blood dynamics. Subsequent sections will delve into specific analytes in saliva and blood, the integration of these biomarkers into clinical decision-making, and the current challenges that must be addressed before routine clinical adoption. By synthesizing current evidence, we highlight both the promise and the remaining gaps aiming to inform a roadmap for translating these technologies into improved patient care.

## 2. Biological Basis of Liquid Biopsies in Oral Oncology

### 2.1. Tumor Biology of HPV-Associated Oral Cancer

HPV-driven cancers follow a distinct pathway from tobacco-induced carcinogenesis. High-risk HPV16 integrates into host epithelial cells, with viral E6/E7 oncoproteins inactivating p53 and Rb. Unlike HPV(−) tumors, which harbor TP53 and CDKN2A loss, HPV(+) tumors usually retain wild-type p53 and p16, instead showing PIK3CA mutations, TRAF3 loss, or E2F1 amplification [8] [Figure 1]. Viral DNA and integration sites serve as unique tumor markers, informing liquid biopsy approaches. Field cancerization is typical of tobacco use, whereas HPV promotes multifocal infection; synchronous or metachronous HPV(+) lesions occur but are uncommon, with persistent local infection driving progression [1].

While high-risk HPV infection is a necessary initiating event, it is not sufficient for malignant transformation. The majority of individuals with persistent oral HPV infection do not develop oral or oropharyngeal cancer, highlighting the limited specificity of HPV positivity as a predictive marker. Progression to malignancy depends on multiple additional factors, including viral integration, onco-protein expression, immune evasion, and host genetic susceptibility, all of which contribute to the biological heterogeneity observed in HPV-driven carcinogenesis [15].

### 2.2. Shedding of Tumor Biomarkers into Biofluids

Solid tumors release debris into saliva and blood, representing local and systemic compartments. In the oral cavity, saliva is in direct contact with tumor surfaces, carrying exfoliated cells and tumor-derived cfDNA, RNA, and proteins, including HPV DNA in HPV-driven OPSCC. Oral rinses often detect HPV16 DNA, reflecting both viral genomes and human tumor alterations. A landmark 2015 study demonstrated that when both saliva and plasma were tested in head and neck cancer patients, tumor DNA was detectable in 96% of cases and importantly, some patients had detectable tumor DNA only in saliva but not in plasma (particularly for oral cavity tumors) [16]. This underscores that saliva is a critical medium for capturing localized tumor signals that may not disseminate widely.

### 2.3. Shedding of Tumor Biomarkers into Blood

Bloodstream, on the other hand, is a highway for systemic dissemination. Tumor cells can invade local vasculature, releasing circulating tumor cells (CTCs) and microscopic emboli, and tumor cell death (via apoptosis/necrosis or immune attack) releases fragments of DNA (circulating cell-free DNA) into circulation. In HPV-associated cancers, the viral oncoproteins do not directly cause a unique pattern of dissemination, but the presence of viral DNA in tumor cells provides a highly specific marker in blood [17].

Beyond DNA, tumor cells release RNA molecules (mRNAs, microRNAs, long non-coding RNAs) and proteins into circulation and saliva. Exosomes (30–150 nm vesicles) are particularly interesting; these tumor-derived vesicles can carry HPV oncogene transcripts or even portions of HPV DNA enclosed in a lipid membrane. There is evidence that HPV16 DNA and E6/E7 RNA are present in exosomes from both plasma and saliva of HPV+ OPSCC patients [18]. Additionally, the host’s immune response provides circulating antibodies (IgG) against HPV16 E6 and E7 oncoproteins in most patients with HPV(+) OPSCC; these antibodies are readily detected in serum and persist over time [18]. Collectively, these analytes (DNA, RNA, protein, antibodies) form a multi-dimensional biomarker pool accessible via minimally invasive means.

### 2.4. Saliva vs. Blood Dynamics

The biodistribution of tumor-derived analytes differs between saliva and blood, conferring each fluid certain advantages. Saliva is ideal for sampling the primary tumor microenvironment. Especially for tumors confined to the oral cavity/oropharynx mucosa, a high proportion of tumor DNA may shed outward into saliva rather than into the bloodstream. For example, in oral cavity cancers, salivary ctDNA often mirrors tumor mutations with high allele fractions [19]. In an analysis stratifying by site, salivary tumor DNA detection rates were highest for oral cavity cancers and lower for oropharyngeal cancers, whereas plasma burden showed the opposite trend [16]. Specifically, in HPV-associated OPSCC, one study found HPV DNA was detectable in saliva of only ~40% of patients but in plasma of ~86% of patients (when using sensitive digital PCR) [16]. This likely reflects that OPSCC tumors often reside in lymphoid tissue with a rich blood supply (facilitating systemic shedding) but sometimes recessed from the saliva stream.

Plasma, conversely, is the medium of choice for capturing systemic and metastatic spread. Plasma is typically favored over serum for cfDNA/ctDNA analysis, as it minimizes leukocyte-derived DNA contamination [20]. Plasma ctDNA analysis has revolutionized investigation of other cancers (e.g., detecting EGFR mutations in lung cancer [21], monitoring residual disease in colon cancer [22]). In HPV+ head and neck cancer, plasma is uniquely valuable for post-treatment surveillance: any viable tumor (whether at the primary site or in distant metastases) can shed HPV DNA into blood, which serves as a whole-body “radar.” Indeed, studies show that a rising ctHPV DNA level in plasma is often the first sign of recurrence, months before a lesion becomes apparent on PET/CT [23]. That said, plasma dilutes tumor markers into the entire blood volume, so extremely early lesions or low tumor burden might fall below the threshold of detection limits [24]. For purely localized (non-invasive) dysplasia, plasma ctDNA could be negative while saliva might still pick up shed HPV copies in the local vicinity. An emerging understanding is that saliva excels at detecting locoregional disease, whereas plasma excels at detecting regional and distant disease spread. These fluids are therefore complementary: for example, in one cohort, combining saliva and plasma testing achieved 96% detection of known tumors, whereas a single sample i.e., saliva or plasma would not have detected some cases [16].

In summary, the biological basis for liquid biopsy in oral oncology rests on three pillars: (1) the unique viral and tumor markers produced by HPV-driven cancers (viral DNA, viral transcripts, distinctive mutation profiles) that are absent in healthy individuals; (2) the continuous shedding of these markers into accessible body fluids (especially saliva for local shedding and blood for systemic spread); and (3) the technological ability to capture and amplify these biomarkers with high sensitivity (using advanced PCR, sequencing, and nanoscale detection methods). With these foundations, we can now explore the specific saliva-derived and blood-based biomarkers under investigation, and how they are being integrated into precision medicine paradigms.

## 3. Saliva-Derived Biomarkers

For HPV-driven oral cancers, saliva is a particularly attractive diagnostic medium given the anatomic proximity of the primary tumor to the oral cavity. In this section, we review major classes of salivary biomarkers and their performance in detecting or predicting HPV-associated oral/oropharyngeal malignancies.

### 3.1. Salivary DNA Biomarkers: Host Tumor Mutations and ctHPV DNA

Saliva contains both host tumor DNA and viral (HPV) DNA, offering complementary signals for HPV-related oral/oropharyngeal cancer. Tumor-derived mutations (e.g., TP53, PIK3CA, FAT1, NOTCH1) are detectable in matched saliva from oral cavity squamous cell carcinoma (OSCC) with high tissue–saliva concordance (>70–80%), and ultra-deep, error-corrected sequencing improves detection of low-frequency variants and subclones [16,17,19,25]. In early OSCC, saliva outperformed plasma (100% vs. 50% mutation detection in a 2015 proof-of-concept), and a meta-analysis estimated ~72% sensitivity for salivary tumor DNA, rising >90% when combined with plasma DNA [16,19]. Although HPV-positive OPSCC has a lower mutational burden, hotspot PIK3CA mutations can be captured in saliva with advanced panels [26].

For viral ctHPV DNA, population screening is limited by the low prevalence of persistent oral high-risk HPV (~1%), but in at-risk or symptomatic cohorts, salivary HPV DNA testing shows high specificity (~94–100%) with moderate sensitivity (~68–96%), depending on assay and setting (qPCR/ddPCR/NGS) [27,28,29]. ddPCR enables single-copy-level detection of plasma ctHPV DNA further boosts diagnostic yield (≈95% sensitivity when either is positive) [30]. Quantitative salivary HPV16 copy number correlates with tumor burden, and persistence after therapy signals elevated recurrence risk, supporting use of serial saliva testing as an adjunct to imaging and plasma ctHPV DNA during surveillance [23,31].

### 3.2. Host–Virus Hybrid Markers (Integration Sites)

When HPV integrates into the host genome, the junction fragments (viral-human chimeric DNA) are unique to each tumor clone. These fragments can theoretically serve as ultra-specific biomarkers, a “molecular tag” of that tumor. Next-generation sequencing can map integration breakpoints in the tumor, and then PCR assays can be designed to detect those specific junctions in saliva or blood. A pilot study in cervical cancer found that patient-specific HPV–human DNA junctions were detectable in serum cfDNA among ~40% of patients, but this was less sensitive than broad HPV DNA tests [32]. In head and neck cancer, this approach is still exploratory. The benefit of integration-specific assays is specificity, essentially zero false positives. But the drawback is that each patient requires a custom assay (personalized medicine approach), and some HPV(+) tumors are episomal (no integration) or have multiple integration sites making it complex. As an early example, Chera et al. noted that their NGS-based ctHPV test (which in part identifies HPV whole-genome fragments potentially including breakpoints) had excellent specificity but occasionally missed tumors that simple E6/E7 PCR could catch [13]. At present, salivary assays targeting integration are not routine, but research in this area could eventually yield “universal” surrogate markers (e.g., detection of HPV–TERT integrations found frequently in tumors).

### 3.3. Transcriptomic Analytes (Cell-Free RNA and MicroRNAs)

Saliva contains both mRNAs and small RNAs (microRNAs, etc.) that can be of tumor origin. RNA is less stable than DNA, but saliva’s RNase activity is lower than that of blood, and saliva collection devices often include RNA-stabilizing buffers [19]. Several studies have profiled salivary mRNAs in OSCC patients. Notably, seven mRNA transcripts (IL8, SAT, S100P, IL1B, OAZ1, DUSP1, HA3) were found elevated in OSCC saliva in an initial study (91% sensitivity/specificity) and validated in a larger cohort (n = 300+) [19]. These transcripts are not HPV-specific, but they reflect tumor presence. For HPV-related cases, one might expect certain viral transcripts (E6/E7 mRNA) or immune response genes to be enriched. Indeed, recent work found that HPV16 E6/E7 mRNA was detectable in saliva rinses of OPSCC patients and correlated with tumor p16 status [10]. However, direct saliva mRNA testing for HPV is technically challenging due to low copy numbers. A potentially richer vein is salivary microRNAs. Saliva from patients with HPV(+) OPSCC showed higher levels of miR-363 and miR-33, according to one analysis [33,34]. Another study reported that miR-127-3p and miR-363 were strongly associated with oral HPV status and could distinguish HPV-positive vs. negative tumors [33,35]. Beyond individual miRs, machine learning classifiers based on salivary microRNA panels have been explored. For example, a panel of 4 salivary microRNAs could differentiate OSCC patients from controls with ~85% accuracy in a small trial. For HPV+ cases, research is ongoing to identify a signature candidates include miR-9, miR-155, miR-200a, and others modulated by the E6/E7 pathway [19,34]. It’s worth noting that salivary RNAs might partly originate from oral inflammatory cells or microbiota; rigorous controls are needed to ensure changes are tumor specific. Regardless, transcriptomic markers add another layer of information (reflecting active gene expression rather than just presence of DNA).

### 3.4. Epigenomic Markers (DNA Methylation)

DNA methylation is pervasive in HNSCC and measurable in saliva. Panels detecting promoter hypermethylation (e.g., EDNRB, KIF1A, NID2, HOXA9) show moderate tissue–saliva concordance (κ ≈ 0.6) and distinguish cases from controls; multi-gene classifiers (e.g., p16, DAPK1, MGMT, CCNA1) also predict recurrence and survival, suggesting a higher salivary “methylation load” reflects aggressive biology [19]. In HPV-associated disease, host (e.g., TFAP2A) and viral regulatory-region methylation including elevated HPV16 L1 CpG methylation in saliva correlate with OPSCC versus asymptomatic carriers. Further large-scale methylome analyses are warranted to identify robust salivary epigenetic signatures.

### 3.5. Proteomic and Metabolomic Biomarkers

Salivary proteomics identifies diagnostic panels enriched for acute-phase, enzymatic, and structural proteins; combinations such as IL-8 plus defensins show case–control discrimination [19]. Direct detection of HPV oncoproteins remains difficult, though a nano-immunoassay chip reported rapid HPV16 E6 detection in saliva (~100% sensitivity; ~88% specificity vs. laboratory PCR), supporting chair-side screening potential pending external validation [36]. Metabolomics consistently shows elevated polyamines (putrescine, spermidine, spermine) in oral cancer higher in invasive than in situ disease along with shifts in amino acids, lactate, and glycolytic intermediates, reflecting altered tumor metabolism [37]. While not HPV-specific, integrating proteomic/metabolomic signatures with viral markers (e.g., saliva or plasma ctHPV DNA) may improve risk stratification and early lesion detection.

### 3.6. Extracellular Vesicles (EVs) and Exosome Cargo

Saliva contains abundant EVs secreted by salivary glands and lining cells. Tumor cells also shed EVs that end up in saliva. These exosomes protect their cargo from degradation and can enrich tumor-specific contents. For instance, an *acoustofluidic exosome isolation* device was used to concentrate salivary exosomes from OPSCC patients, leading to a 15-fold increase in yield of exosomal RNAs and improving HPV16 detection to 80% of confirmed cases [38]. Exosomal content analysis has revealed differences between HPV+ and HPV− HNSCC: one study found that HPV+ cell-derived exosomes contained viral oncogene transcripts and a distinct protein profile (e.g., presence of p16, E6/E7, and immunosuppressive proteins like PTPN11) [39,40]. Salivary exosome proteomics by Fontana et al. identified hundreds of proteins whose levels distinguished oral cancer patients with and without lymph node metastasis, pointing to EV cargo as a source of biomarkers for tumor staging [41]. Exosomes may capture functional aspects of the tumor (proteins/microRNAs influencing invasion or immune evasion) that DNA alone does not convey [42]. However, in OPSCC patients, plasma cfDNA achieved far higher sensitivity for HPV detection than EV-associated DNA (91% vs. 42%; *p* < 0.001) and cfRNA similarly outperformed EV-RNA (83% vs. 50%; *p* = 0.0019) [43]. The rationale is that exosomes may capture functional aspects of the tumor (proteins/microRNAs influencing invasion or immune evasion) that DNA alone does not convey. Figure 2 summarizes the diverse categories of saliva-based biomarkers identified in HPV-associated oral and oropharyngeal cancers.

### 3.7. Diagnostic and Prognostic Performance

Single salivary analytes seldom achieve durable accuracy across settings, with sensitivity limited at low tumor burden and specificity vulnerable to inflammatory noise. By contrast, multi-analyte, saliva-based multi-omics models that fuse ctHPV DNA with host features (targeted methylation loci, microRNA/mRNA panels, and selected metabolites) have potential to outperform single markers. A study reported that pretreatment HPV qPCR achieved 76% sensitivity when both biofluids were assessed, relative to 52.8% for saliva and 67.3% for plasma individually [44]. Additionally, posttreatment dual-compartment liquid biopsy demonstrated 69.5% sensitivity and 90.7% specificity for predicting recurrence within 3 years. These findings were later confirmed in a prospective study, which yielded 65% sensitivity and 87% specificity [23,45]. Non-invasive collection supports serial monitoring in high-risk individuals and survivors, and emerging methods (microfluidic enrichment, ultra-deep sequencing) are raising analytical sensitivity. The field’s goal is to be a clinically deployable panel for screening and post-treatment surveillance, which will require large prospective validation, standardized pre-analytics, and regulatory clearance. Current evidence already establishes saliva as a valuable liquid-biopsy matrix in oral oncology.

## 4. Blood-Based Biomarkers

Peripheral blood is the most widely studied liquid biopsy medium in oncology. In the context of HPV-associated oral cancers, blood-based biomarkers offer a systemic lens on tumor presence and behavior, complementing the locoregional insights from saliva. Here we discuss key categories of blood-derived biomarkers: circulating tumor DNA, circulating tumor cells, circulating RNAs, extracellular vesicles, tumor-educated platelets, and serologic (immune) markers.

### 4.1. Circulating Tumor HPV DNA (ctHPV DNA) in Plasma

Plasma circulating tumor HPV DNA (ctHPV DNA) is a tumor-specific biomarker detectable by digital PCR or targeted sequencing, with high diagnostic and surveillance accuracy in HPV-positive oropharyngeal cancer [13,17]. Pretreatment detection rates approach ~95% in HPV16-driven disease, and in a 2020 prospective trial two consecutive positives yielded PPV ~94% for recurrence with negative predictive value (NPV) 100% for persistently negative results [13]. ctHPV DNA levels track tumor burden typically falling to undetectable levels after effective therapy and often anticipate relapse by ~4–7 months (median lead time) [13]. Assays usually target E6/E7 by PCR; HPV-targeted NGS can capture multiple genomic fragments and rare types (e.g., CaptHPV) [29], while tumor-tissue–modified viral DNA (TTMV) incorporating methylation features may further enhance specificity for tumor-derived HPV DNA [46]. Baseline plasma ctHPV-DNA identified HPV-associated OPSCC with 89% sensitivity and 97% specificity, and pretreatment levels correlated with tumor burden and HPV integration status [24,47]. Overall, plasma ctHPV DNA has promising potential for clinical application. It is already being used in clinical trials to guide de-escalation (e.g., suspending neck dissection in those who clear ctDNA quickly after chemoRT) [48] and intensification (early adjuvant therapy for molecular residual disease). Regulatory approval may not be far off. A key consideration is that HPV ctDNA positivity is essentially 100% specific for cancer, but false negatives can occur in very small tumors or those anatomically isolated and remote from circulation.

Despite promising sensitivity, false-negative results remain a recognized limitation in liquid biopsy for HPV-associated cancers. Several factors contribute to this issue, including low circulating viral DNA levels in early-stage disease, anatomical and temporal variation in biomarker shedding, and technical variability across detection platforms. For example, low tumor burden or minimal residual disease may fall below assay detection thresholds, particularly in salivary sampling, which can be more variable than plasma-based detection. Assay-related limitations such as suboptimal sensitivity, pre-analytical variability, and platform-specific differences further impact reliability. A recent prospective study underscores that even optimized ctHPV DNA assays can yield false negatives, especially in early-stage or low-volume tumors [13].

### 4.2. Circulating Tumor Cells (CTCs)

CTCs are intact cancer cells that detach from the primary tumor or metastases and enter the bloodstream. They are exceedingly rare (often just 1–10 CTCs per 10 mL blood in solid tumors) but can be enriched and identified by various technologies [49]. In HNSCC, CTC research has been less prolific than in breast or prostate cancer, but a few studies provide insight [50]. HPV-positive HNSCC sheds CTCs that can express p16^INK4A^ and/or HPV16 E6/E7 transcripts, enabling confirmation of viral origin. In a flow-cytometry cohort (n = 41), p16-positive CTCs were detected in ~51%; notably, some patients had HPV DNA–positive tumors despite p16-negative IHC, indicating that CTC profiling can resolve tissue–marker discordance [17]. Using microfluidic capture with RNA-ISH, Economopoulou et al. detected E6/E7-positive CTCs in 30% pre-treatment and 44% post-treatment, exclusively among HPV16-positive cases (0% in HPV-negative) [51]. CTC burden appears prognostic: ≥3 CTCs/mL correlated with worse progression-free survival [52], while p16-positive CTCs are associated with better outcomes on multivariable analysis [17] potentially reflecting radiosensitive phenotypes versus p16-negative/mesenchymal, treatment-resistant clones. Routine monitoring remains impractical due to rarity and technical demands, but CTCs offer unique material for functional assays and clonal genomics and may complement ctDNA (discordance can suggest cell death vs. active dissemination).

### 4.3. Cell-Free Tumor DNA (Tumor Mutations & Methylation in Plasma)

Beyond viral DNA, HPV-associated tumors, like all cancers, release fragments of human tumor DNA into plasma. These fragments can be detected by looking for tumor-specific mutations or copy number aberrations. For HPV(−) HNSCC, this is the main approach (e.g., TP53 mutations in plasma). In HPV(+) cases, viral DNA is the simplest marker but combining it with human tumor DNA analysis could enhance sensitivity or provide additional information (like presence of certain mutations that might guide therapy). For instance, about 20–30% of HPV(+) HNSCC have mutations in PIK3CA [53]; detecting a PIK3CA mutation in plasma ctDNA alongside HPV16 might indicate a high tumor fraction and possibly identify a targetable pathway (PI3K inhibitors). A 2021 systematic review found that in studies which attempted both, plasma HPV DNA was more frequently detected than somatic mutations in HPV(+) cases (since not all HPV+ tumors have common mutations) [17]. However, tumor DNA methods like low-coverage whole-genome sequencing for copy number instability (CNI) can be applied universally. Schirmer et al. developed a “CNI score” from shallow sequencing of plasma cfDNA; it distinguished HNSCC patients from controls with AUC ~ 0.87 (95% CI 0.79–0.93) [54]. High CNI scores correlated with advanced stage and worse survival. This approach does not rely on knowing HPV status or specific mutations it’s a genome-wide measurement of chromosomal chaos in ctDNA. Another study by Rapado-González et al. found that at least one somatic mutation was identified in plasma in 37.5% of HNSCC patients (with deep sequencing of a gene panel), and detection strongly associated with higher tumor stage. For HPV+ OPSCC, some researchers have noted that combining TP53 mutation assays (for HPV− tumors) with HPV DNA assays (for HPV+) yields a comprehensive liquid biopsy covering both etiologies, as these are largely mutually exclusive markers [55].

Plasma ctDNA methylation provides an orthogonal, tissue-linked signal. Methylation-only assays for head–neck cancers achieve high specificity (~98%) but modest sensitivity (~40–50%). In HPV-positive OPSCC, combining HPV DNA with host-gene methylation improves attribution to tumor-derived fragments versus benign oral HPV shedding [56]; recurrent loci (e.g., ZNF662, KLF8) serve as stable fingerprints. In practice, HPV ctDNA alone is usually sufficient when HPV status is known, whereas mutation + methylation panels add value for screening (unknown HPV status) and for increasing specificity in MRD assessment. Researchers are experimenting with algorithms that integrate ctDNA mutations, and clinical factors to stratify patients [57].

### 4.4. Circulating Cell-Free RNA (cfRNA)

Tumor-derived mRNA and non-coding RNAs enter plasma largely within EVs or protein complexes, conferring nuclease resistance. Direct tumor mRNA detection in HNSCC plasma is not routine, but HPV E7 transcripts have been identified in serum exosomes in cervical cancer, supporting feasibility for EV-based E6/E7 assays [58]. Plasma microRNAs are more tractable: profiles differ between HPV-positive and HPV-negative HNSCC (e.g., miR-9, miR-20b, miR-34a) [59], and exosomal miR-27b-3p, miR-491-5p, miR-630, and miR-1910-5p are dysregulated in locally advanced disease, with longitudinal miR-491-5p changes associating with survival [17]. These microRNA signatures are not HPV-specific but can augment detection and risk stratification. Tumor-educated platelet RNA and cytokine panels remain exploratory in HNSCC and require prospective validation.

### 4.5. Extracellular Vesicles in Blood

Plasma carries abundant tumor-derived EVs, but isolating tumor-specific vesicles from a large background remains challenging. Immunocapture (e.g., EpCAM, PD-L1) enriches relevant EVs. In HNSCC, exosomal PD-L1 decreases after effective therapy and rises with recurrence, and patients with active disease harbor more PD-L1+ EVs than those with no evidence of disease supporting EV-based immune monitoring [60,61]. Given the immune-rich microenvironment of HPV+ tumors, PD-L1 and related immune EV markers may refine risk stratification and predict checkpoint response. EV proteomics has nominated candidates (e.g., THBS2, EGFR) that distinguish cases from controls, but findings are preliminary and require external validation [17].

### 4.6. Serologic Immune Biomarkers

One hallmark of HPV-driven cancers is the presence of circulating antibodies against HPV early proteins. Approximately 65–85% of patients with HPV+ OPSCC have serum antibodies to HPV16 E6 and/or E7 at diagnosis [1]. These antibody levels are exceedingly rare in cancer-free individuals (<1% prevalence for HPV16 E6 in the general population) [18], making them a highly specific marker of an HPV-induced malignancy (though not telling where the tumor is located). Prospective cohorts (like the PLCO study) showed that HPV16 E6 seropositivity can precede OPSCC by decades and confers a massive increase in risk (E6 seropositive individuals had ~25% risk of developing OPSCC within 10 years vs. ~0.1% background risk) [18]. This suggests serum antibody screening could identify high-risk persons for closer surveillance. Clinically, if a patient has neck nodes and is HPV16-E6 seropositive, it strongly points to an occult OPSCC primary (one study reported >95% PPV in that scenario) [10]. It’s important to note that antibody signals are more delayed and less dynamic than ctDNA. They may take months to rise or fall. But they could provide orthogonal evidence of recurrence or serve as a screening filter. There is interest in whether circulating T cells specific to HPV antigens could indicate an ongoing anti-tumor immune response; early data suggests HPV-specific T cells can be found in the blood, but their correlation with tumor control is unclear. Figure 3 correlates circulating biomarkers in blood and their corresponding detection technologies for HPV-induced cancers.

## 5. Integration into Precision Medicine Paradigms

The advent of saliva and blood liquid biopsies for oral cancer comes at a time when oncology is increasingly embracing precision medicine, tailoring interventions to the molecular characteristics of each patient’s disease and adjusting management in real-time based on dynamic biomarkers. In this section, we discuss how liquid biopsy biomarkers can be integrated into precision medicine paradigms in HPV-driven oral oncology, including companion diagnostics, adaptive therapy strategies, point-of-care technologies, and public health applications.

### 5.1. Companion Diagnostics and Therapeutic Decision-Making

Although there are no drugs that target HPV oncoproteins directly, plasma ctDNA profiling enables indirect precision matching in HPV-positive HNSCC: detection of actionable co-alterations (e.g., PIK3CA) can support PI3K-pathway trials when tissue is limited; emerging resistance variants (e.g., NOTCH1) can prompt radiosensitizer use or systemic switches; and mutations such as CTNNB1 may stratify response to immunotherapy [17]. A baseline liquid-biopsy panel at diagnosis can confirm HPV status and uncover co-drivers (e.g., PTEN loss), functioning as a companion diagnostic for trial enrollment and therapy selection.

HPV-positive tumors are highly radiosensitive, enabling trials of treatment de-escalation to reduce toxicity. Plasma ctHPV DNA clearance during therapy identifies patients suitable for dose/drug reduction: in a prospective study (NCT03515538), mid-treatment clearance correlated with excellent outcomes on de-intensified regimens, whereas persistent ctDNA flagged patients needing standard or intensified therapy. Post-treatment MRD positivity (ctHPV DNA detectable) can trigger closer surveillance or pre-emptive adjuvant interventions. This risk-adapted, ctDNA-guided approach mirrors strategies in colorectal and lung cancer and is now being operationalized in head and neck trials.

### 5.2. Digital Health and Point-of-Care (POC) Devices

Clinical integration hinges on rapid, low-burden testing. Microfluidic POC cartridges can detect HPV16 DNA in unprocessed saliva via isothermal amplification in ~30 min with PCR-comparable accuracy [36]. Another is the acoustofluidic exosome isolator from UCLA/Duke, which could be coupled with a downstream readout to form a “spit test” for OPC screening. These devices are often small, portable cartridges that can be run in a clinic setting or potentially even as home tests. Coupling cartridges to smartphone readers with on-device machine learning (ML) that analyzes real-time amplification curves further improves signal-to-noise in field settings [62]. These tools could decentralize screening to dental clinics and pharmacies and eventually homes pending external validation, standardized QC, and secure result reporting.

### 5.3. Cost-Effectiveness and Health Equity

Equitable precision requires assays that are affordable, scalable, and easy to deliver. Although ddPCR/NGS currently carry higher per-test costs, unit prices may fall with volume and multiplexing; saliva HPV DNA testing can be implemented at low cost and without phlebotomy. Health-economic modeling from the UK suggests that ctHPV DNA–guided surveillance may reduce follow-up costs by decreasing imaging frequency and facilitating earlier, potentially less invasive interventions, though these findings remain model-based and require clinical validation [63]. Another modeling study suggested that screening high-risk populations (such as men with history of multiple oral sex partners) with an annual saliva HPV test might be cost-effective if the test sensitivity and compliance are adequate, because the cost of treating late-stage OPSCC is very high. Importantly, saliva collection itself is non-invasive and culturally acceptable in most settings. It does not require phlebotomy or a clinical professional. This could allow outreach programs in low-resource or rural communities: for example, mail-in saliva kits for HPV testing, like colorectal cancer stool tests. People could collect a saliva sample at home and send it to a lab for analysis, eliminating travel barriers. Such strategies could improve early diagnosis in populations that currently present with advanced disease due to lack of access and poor compliance.

Biomarker performance can vary by demography and context (e.g., oral microbiome, tobacco/betel-quid exposure, care access), so algorithms require prospective, multi-site validation with prespecified subgroup analyses (race/ethnicity, sex, age) and fairness metrics plus site-level calibration. Community dental clinics are high-yield access points for saliva testing, reaching people who are rarely seen in primary care or ENT (Otorhinolaryngology); embedding brief screening during routine hygiene visits could trigger earlier referral than symptom-driven presentation (e.g., neck mass). Implementation should include standardized pre-analytics, multilingual materials, low out-of-pocket costs, and clear navigation pathways for positive screens (confirmatory testing, expedited specialty care). Dentists and dental hygienists might play a key role in implementing saliva-based screening as part of routine oral health assessments, as they are often the only health professionals examining the mouth and throat area regularly.

### 5.4. Precision Surveillance and Personalized Screening

Risk-adapted follow-up protocols may offer advantages over fixed schedules and warrant further evaluation. After curative therapy for OPSCC, patients with serially negative plasma ctHPV DNA and saliva HPV tests can be seen less frequently with fewer body/organ scans, reducing the burden of hospital visits without compromising safety. Conversely, low-level or rising ctHPV DNA may warrant consideration of shortened follow-up intervals and targeted imaging, pending further validation. While this risk-adapted model remains investigational, its implementation would require prespecified thresholds (e.g., confirmation on a second draw), integration with clinical and radiologic assessment, and periodic outcome audits. Such an approach could eventually support more personalized surveillance strategies compared to fixed-interval follow-up of 2–3 months.

### 5.5. Regulatory and Implementation Considerations

To integrate into standard care, liquid biopsy tests must gain regulatory approval (FDA in U.S., CE mark in Europe, etc.) and demonstrate clinical utility. Incorporation into clinical guidelines (e.g., NCCN or ASCO guidelines) will follow once prospective trials report improved outcomes with ctDNA-guided interventions [13]. From an implementation standpoint, hospital laboratories may need to adopt new workflows (for example, setting up in-house ddPCR for HPV, or sending samples to specialized labs until in-house capacity is built). Oncologists and surgeons will need education on interpreting liquid biopsy results. For instance, understanding false-positive rates and not overreacting to a single low-positive, but confirming with a second test (since sequential positivity had near 100% PPV in studies).

In conclusion, integrating liquid biopsies into precision medicine means using the right test, at the right time, for the right patient, to make better decisions. HPV-driven oral cancers present a compelling opportunity: the biology provides clear targets (viral DNA) and the clinical needs are evident (early detection and tailored surveillance). As evidence gathers, we envision a future where a patient’s journey from diagnosis to survivorship is guided by a suite of minimally invasive tests—a saliva sample that helps diagnose the tumor and define its viral etiology, a blood test that monitors molecular remission, and digital health tools that deliver results quickly to clinicians and patients. This integration will transform the care pathway into one that is more proactive, personalized, and patient-friendly.

## 6. Current Challenges and Knowledge Gaps

Despite the excitement surrounding liquid biopsies in oral cancer, significant challenges must be addressed before these tools can be fully mainstreamed. Some challenges are technical, some biological, and others logistical or regulatory. We outline the key hurdles and knowledge gaps below:

### 6.1. Pre-Analytical Variability

The accuracy of liquid biopsy assays heavily depends on standardized sample collection and processing [Table 2]. Saliva samples can vary with time of day, recent eating or oral hygiene, and collection method (passive drool vs. oral rinse vs. swab) [19]. Inconsistent protocols lead to differences in biomarker yield—for example, an aggressive gargle might dislodge more tumor cells than a light spit. Similarly, blood samples for ctDNA require careful handling: plasma should be separated promptly (within hours) or collected in special tubes to prevent white cell lysis and dilution of tumor DNA with genomic DNA. A lack of uniform pre-analytics can cause class imbalance in data studies may overestimate performance if high-quality samples are used (case) and lower quality for controls, or vice versa. One knowledge gap is understanding how factors like oral inflammation or periodontal disease affect salivary biomarkers (they might elevate certain DNA or protein markers, causing false positives). We also lack consensus on thresholds for calling a ctHPV DNA test positive low-level readings can occur from tagging errors or transient benign HPV presence. Thus, establishing robust cutoff criteria (perhaps requiring two consecutive positive results to define molecular recurrence) is crucial [13].

### 6.2. Sensitivity vs. Specificity Trade-Offs

Many current assays face a trade-off between sensitivity and specificity. Ultra-sensitive methods can detect vanishingly small signals, but at the risk of false positives (contamination or off-target amplification). For instance, sequencing-based mutation detection can pick up “tumor-like” mutations that are Clonal hematopoiesis of indeterminate potential (CHIP) mutations from white blood cells. In older patients, CHIP is common and could yield false mutation calls in plasma cfDNA. Solutions like matched white blood cell sequencing can mitigate this but add cost and complexity. For HPV DNA tests, specificity is generally high, but one scenario could cause false positives: oral HPV infection in the absence of cancer. A person who acquired HPV16 recently might shed viral DNA in saliva for months and then a saliva HPV test could be positive, though no cancer is present (the infection may clear). This scenario is akin to low positive predictive value when prevalence is low. How do we distinguish an “HPV carrier” from someone with a developing malignancy? This is a major knowledge gap. Potentially, quantitative viral load or concurrent host biomarkers (like cyclin E overexpression) could help, but it remains unsolved.

### 6.3. Tumor Heterogeneity and Evolution

Liquid biopsy assays often target one or a few markers (e.g., HPV16 DNA). If a tumor is heterogeneous say, mostly HPV-driven but with a minor clone that is HPV-negative a sole-focus test could miss signals. Moreover, as tumors evolve (especially under treatment pressure), their shedding profile might change. An HPV-positive tumor that loses the HPV plasmid (extremely rare, but hypothetically) or substantially downregulates viral gene expression might shed less viral DNA even while progressing leading to a false sense of security. We need more data on longitudinal kinetics: how do salivary and plasma markers change from pre-treatment to during therapy to remission or recurrence? Early studies suggest that effective treatment causes an exponential drop in ctHPV DNA within days to weeks [13], but the exact decay kinetics and the possibility of late re-emergence (blips) need elucidation. Additionally, the presence of anatomical subsites with differing drainage (a tongue base tumor might shed to saliva differently than a tonsil tumor) means we might need anatomical site-specific studies with detailed attention to the interpretation of results.

### 6.4. False Negatives and “Occult” Disease

While ctDNA and HPV assays are sensitive, they can yield false negatives, especially for small tumors. A significant gap is how to handle a patient who has clinical risk factors but negative liquid biopsy. For example, a heavy smoker with a neck mass: if plasma HPV is negative, one might assume HPV-negative cancer but if a node biopsy shows HPV-positive tumor, that’s a false negative plasma result (perhaps due to low volume or node encapsulation). Thus, liquid biopsies should augment, not replace, tissue diagnosis when possible. In surveillance, a recurring theme is what if imaging is clean but ctDNA is positive (probably treat as recurrence), vs. imaging suspicious but ctDNA negative (possibly a false imaging finding or a non-biologically significant lesion). The latter is tricky; might one observe longer if ctDNA is clean? These scenarios require prospective validation to avoid harm from misinterpreting a false negative or false positive.

### 6.5. Data Interpretation and Bioinformatics

The analysis of liquid biopsy data is complex. Sequencing data requires filtering, alignment, and variant calling with distinction between tumor DNA and noise. Standardizing bioinformatics pipelines is needed to avoid “batch effects” or overfitting. For example, different groups may report different “sensitivity” for salivary mutations partly because of differences in how strictly they define a positive call. To compare studies and combine data, common frameworks and reference standards (like reference HPV DNA “spike-in” controls in saliva) are needed. There’s also the challenge of multi-omic data integration. If one collects thousands of data points (mutations, methylation signals, RNA levels, etc.) on a small patient sample, one risks discovering patterns that are not generalizable (a classic curse of dimensionality with small N). Privacy concerns can be addressed via federated learning approaches; an emerging concept where models are trained across multiple institutions’ data without centralizing the data itself.

### 6.6. Regulatory Hurdles

Liquid biopsy assays often start as lab-developed tests (LDTs). Gaining regulatory approval requires demonstrating analytic and clinical validity and clinical utility. For relatively rare outcomes like early OPSCC, proving utility might require large trials which are costly. No FDA-approved saliva test for oral cancer exists yet. For ctHPV DNA, commercial companies are pursuing approval; one test has breakthrough device designation from FDA [64]. Regulatory agencies will scrutinize false positive rates, potential psychological harm of positive screens, and whether use of the test truly improves patient outcomes (e.g., does surveillance ctDNA lead to earlier salvage that improves survival? That hasn’t been definitively proven yet, though it’s strongly implied). Filling this gap will require randomized studies where one arm uses liquid biopsy–guided intervention and the other standard care, to see if outcomes differ.

In sum, while liquid biopsies hold promise, addressing these challenges is essential to avoid pitfalls like overdiagnosis, unnecessary anxiety, or misallocation of resources. Ongoing research is actively tackling many of these issues: large cohorts are being assembled to refine thresholds; new assays (like combined HPV and host markers) aim to reduce false negatives/positives; and international consortia are working on standardizing protocols (e.g., the NCI’s LiqBio-HNC working group; https://prevention.cancer.gov/research-areas/networks-consortia-programs/lbc (accessed on 22 October 2025)). Bridging these gaps will ensure that when liquid biopsy tools enter routine use, they do so safely, effectively, and equitably.

## 7. Future Directions

As we look ahead, the landscape of liquid biopsies in oral cancer is poised to evolve rapidly. Here, we outline key future directions and research avenues that could further advance the field and address current limitations:

### 7.1. Multi-Omics and Machine Learning Integration

One clear trajectory is moving from single-marker tests to multi-omics panels analyzed by sophisticated algorithms. This means simultaneously examining DNA, RNA, proteins, and even metabolic markers in saliva or blood to capture a more comprehensive tumor fingerprint. For instance, a future screening tool might analyze an individual’s saliva for HPV DNA (genomics), specific microRNA and mRNA patterns (transcriptomics), a set of methylated genomic loci (epigenomics), and select protein markers, feeding all that data into an AI-driven model that outputs a risk score for oral cancer. Early efforts in this vein are promising but require larger training datasets. Artificial intelligence (AI) and deep learning can detect subtle patterns that humans cannot; for example, a machine learning model might recognize a combination of ten weak signals (each individually not significant) that together strongly indicate a tumor presence. Federated learning is a particularly relevant approach: institutions could collaboratively train algorithms on multi-omics liquid biopsy data without sharing raw patient data (preserving privacy). This approach can help surmount the data scarcity challenge and avoid bias that comes from training on homogenous populations. A push towards open-source code and models in the liquid biopsy community will also accelerate progress and trust, analogous to the way The Cancer Genome Atlas (TCGA) spurred innovations by making genomic data publicly available.

### 7.2. Liquid Biopsy in Immuno-Oncology

With the rise of immunotherapy for various cancers, including trials in HNSCC, there is interest in using liquid biopsies to guide immunotherapy decisions. Future research will explore if ctDNA dynamics can serve as an early surrogate for immunotherapy response (e.g., a rapid clearance of ctHPV DNA after starting a PD-1 inhibitor might predict durable response, whereas rising ctDNA might precede clinical progression). Additionally, tracking immune repertoire via blood (T-cell receptor sequencing to see if HPV-specific clones expand) or monitoring exosomal PD-L1 could refine patient selection for therapies. For instance, a high burden of exosomes expressing PD-L1 and high ctDNA might suggest a strongly immunosuppressive, active tumor—a candidate for combination immunotherapy or clinical trials, whereas a patient with no ctDNA and residual mass (possible radiation fibrosis rather than tumor) might avoid unnecessary salvage surgery.

### 7.3. “Saliva-on-a-Chip” and At-Home Testing

Technological miniaturization will likely yield robust saliva testing devices that can be used at point-of-care or even mailed to patients’ homes. Like how diabetic patients use home glucometers, oral cancer survivors might use a home saliva HPV test kit at prescribed intervals between clinic visits. One can envision a smartphone attachment that analyzes a saliva droplet for HPV DNA using isothermal amplification and gives a result within 15 min sending data securely to the oncology team. This could particularly benefit patients who live far from cancer centers by reducing travel for routine surveillance. To reach this stage, further engineering is needed to ensure lab-level sensitivity in a portable format and to build fail-safes (e.g., invalid sample detection). But given the trajectory of microfluidics and the successful prototypes already reported, this is a matter of “when” not “if.”

### 7.4. Addressing Unknowns and Broadening Scope

Future research will also need to address currently under-studied areas: HPV-driven malignancies beyond the oropharynx (e.g., some oral tongue cancers, or HPV in sinonasal carcinomas)—can liquid biopsy similarly aid those? Cross-applying knowledge from other solid tumors will be crucial: for example, the concept of “molecular residual disease” detection by ctDNA has taken hold in colorectal and lung cancer; translating those trial designs to oral cancer (where one could consider adjuvant therapy only if ctDNA remains positive post-surgery) is on the horizon. Furthermore, lessons from liquid biopsy in nasopharyngeal carcinoma (where Epstein–Barr Virus (EBV) DNA is used for population screening in Hong Kong with success) provide a blueprint for HPV—large trials could evaluate if screening at-risk populations with saliva HPV DNA reduces late-stage disease (the evidence isn’t there yet and is a target for future study).

In summary, the future of liquid biopsies in oral cancer lies in more comprehensive, intelligent, and accessible applications. Multi-analyte tests interpreted by AI will boost performance; integration with immunotherapy and personalized regimens will maximize clinical benefit; and technology will make testing easier and more equitable. Importantly, all these advancements must be accompanied by rigorous clinical validation to ensure they genuinely improve patient outcomes—the ultimate goal of any future innovation.

## 8. Clinical Implications

Emerging liquid biopsy technologies hold promises to impact on several aspects of oral oncology, ranging from early detection to long-term survivorship care. Below, we outline potential clinical implications for both providers and patients.

### 8.1. Earlier Detection and Diagnosis

Currently, many HPV-positive oropharyngeal cancers present with advanced lymph node metastases (often Stage III/IV) despite small primary tumors [64]. With liquid biopsy tools, there is a realistic hope of migrating diagnosis to earlier stages. For instance, a middle-aged adult with persistent oral HPV infection (detected via a saliva test) and rising HPV16 DNA levels over time could be flagged for endoscopic evaluation, potentially catching a tonsillar carcinoma when it is still in situ or microinvasive. Earlier tumor detection generally leads to better outcomes and de-escalation of treatment intensity. This could mean more patients treated with single-modality surgery or radiation instead of chemoradiation, reducing long-term morbidity (xerostomia, dysphagia, etc.). In essence, liquid biopsy-guided screening could create an “HPV oral Pap smear” equivalent, shifting a proportion of cases from advanced to early, much as cervical Pap screening did for cervical cancer. Furthermore, novel liquid biopsy tests indicate they could accurately detect HPV-associated cancers in asymptomatic individuals many years before they are ever diagnosed with cancer [7].

### 8.2. Enhanced Surveillance and Survivorship Care

Perhaps the most immediate impact will be on post-treatment surveillance protocols. Standard practice relies on scheduled imaging and exams, which are resource-intensive and sometimes anxiety-provoking for patients (“scanxiety”). Introducing plasma ctDNA tests every 3–6 months could improve the efficiency of surveillance: a negative highly sensitive ctDNA result gives reassurance of remission with a very high NPV, potentially allowing fewer imaging studies (unless symptoms arise), whereas a positive result triggers prompt investigation without waiting for routine scan intervals. Patients would benefit from the psychological comfort of “molecular all-clear” blood tests. Over the long term, this might also translate to survival benefits catching recurrences when still localized (since ctDNA often turns positive when tumor burden is low) means more patients eligible for curative salvage (like neck dissection or stereotactic radiotherapy).

### 8.3. Empowerment and Compliance

An often-underappreciated aspect is patient engagement. Liquid biopsy results, especially those delivered through accessible means (like a text “all clear” for a negative test), can engage patients in their care. Seeing objective molecular evidence of remission can reinforce healthy behaviors (smoking cessation, continued follow-up). Conversely, detection of persistent virus or tumor signals might motivate lifestyle changes or adherence to vaccination recommendations for family (although HPV vaccine’s role in secondary prevention is unproven, awareness often spreads within families). Moreover, ease of testing (spitting in a cup vs. scheduling a scan) may improve follow-up compliance, especially in populations that historically have higher loss to follow-up.

### 8.4. Challenges in Implementation

Clinicians will need to integrate these tools thoughtfully. Guidelines and education must be updated so providers know how to interpret results and avoid overreacting to false positives or underreacting to unexpected findings. Tumor boards may include a molecular pathologist or lab director more routinely to assist with interpretation of borderline results. The health system will also have to deal with potential increased workload from earlier interventions (finding more early lesions to treat) but potentially reduced burden from treating fewer late-stage complications.

In conclusion, clinical practice in oral oncology has potential to evolve from a predominantly reactive model focused on treating visible disease and detecting recurrence radiographically toward a more proactive, molecularly informed approach that enables earlier intervention and personalized follow-up. If validated in prospective studies, such strategies have the potential to improve survival, reduce treatment-related morbidity, and enhance quality of life, key goals in contemporary oncology.

While liquid biopsy technologies show promising clinical applications, it is important to interpret the findings of this review considering certain limitations. This review is narrative in nature and does not include formal quality appraisal or risk-of-bias assessment of included studies. The selection of articles was based on topical relevance, recency, and impact, which may introduce selection bias toward larger or more prominent studies. Only English-language, peer-reviewed literature was considered, potentially excluding relevant data published in other languages or formats. As a result, findings should be interpreted as a qualitative synthesis rather than a comprehensive or graded evidence summary.

## 9. Materials and Methods

We conducted a narrative literature review to synthesize recent evidence on liquid biopsy biomarkers in HPV-driven oral and oropharyngeal squamous cell carcinoma. To enhance transparency and reproducibility, we adopted a structured search strategy without any formal characteristics of a systematic review. We conducted a comprehensive literature search to gather pertinent studies on HPV-driven oral/oropharyngeal cancer and liquid biopsy biomarkers. The primary data sources were PubMed/MEDLINE and Embase, with additional queries in Web of Science for cross-referencing. The search timeframe spanned January 2015 to July 2025, chosen to capture contemporary research and technological advances. Key search terms (used in various combinations) included: “HPV OR human papillomavirus”, “oral cancer OR oropharyngeal cancer”, “liquid biopsy”, “saliva biomarkers”, “circulating tumor DNA (ctDNA)”, “circulating HPV DNA”, “circulating tumor cells (CTCs)”, “salivary diagnostics”, “head and neck cancer biomarkers”, “HPV cell-free DNA”, “salivary exosomes”, and “HPV surveillance recurrence”.

We prioritized peer-reviewed, PubMed-indexed studies providing high-level or novel evidence, including meta-analyses, systematic reviews, and large cohort or prospective trials. Seminal earlier studies were also cited when foundational for epidemiologic or mechanistic understanding. Inclusion decisions were based on topical relevance, recency (2015–2025), and contribution to the field, rather than predefined quantitative criteria. Special consideration was given to studies from 2020–2025, reflecting the rapid progress in this period—for instance, multiple prospective trials of ctHPV DNA and new meta-analyses were published in these years. We excluded papers not in English and small case reports unless they illustrated a unique point not covered by larger studies. Extracted information included study type and sample size, biomarker category, analytical method, diagnostic or prognostic performance (e.g., sensitivity, specificity, predictive values, lead time), and principal findings.

Our review synthesis was iterative: as themes emerged (e.g., saliva vs. plasma differences, or assay standardization issues), we conducted focused secondary searches to fill any gaps (such as “HPV saliva microRNA 2022” for the transcriptomics subsection). We explicitly note that no PROSPERO registration, PRISMA flow diagram, or formal risk-of-bias or quality-assessment tools were applied, as this review was not designed as a systematic evidence appraisal. Rather, it provides an integrated overview of current advances and knowledge gaps in the evolving field of HPV-related liquid biopsy research. Through this approach, we aimed to ensure a thorough and up-to-date appraisal of the topic.

## Figures and Tables

**Figure 1 jcm-14-07598-f001:**
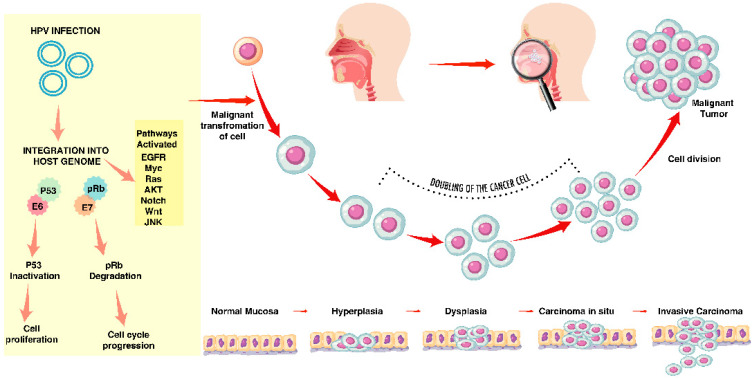
HPV-mediated molecular events and progression to invasive carcinoma. Human papillomavirus (HPV) infection drives carcinogenesis by the integration of viral DNA into the host genome. Viral oncoproteins E6 and E7 inactivate the tumor suppressors p53 and pRb, respectively, causing uncontrolled cell proliferation and cell cycle progression. Activation of several oncogenic signaling pathways, such as EGFR, Myc, Ras, AKT, Notch, Wnt, and JNK. Infected epithelial cells undergo malignant transformation, and this is followed by clonal expansion and progressive genetic and epigenetic changes. Histopathologic progression is illustrated from normal mucosa to hyperplasia, dysplasia, carcinoma in situ, and invasive carcinoma. The scheme illustrates sequential events of HPV-induced tumor initiation and progression.

**Figure 2 jcm-14-07598-f002:**
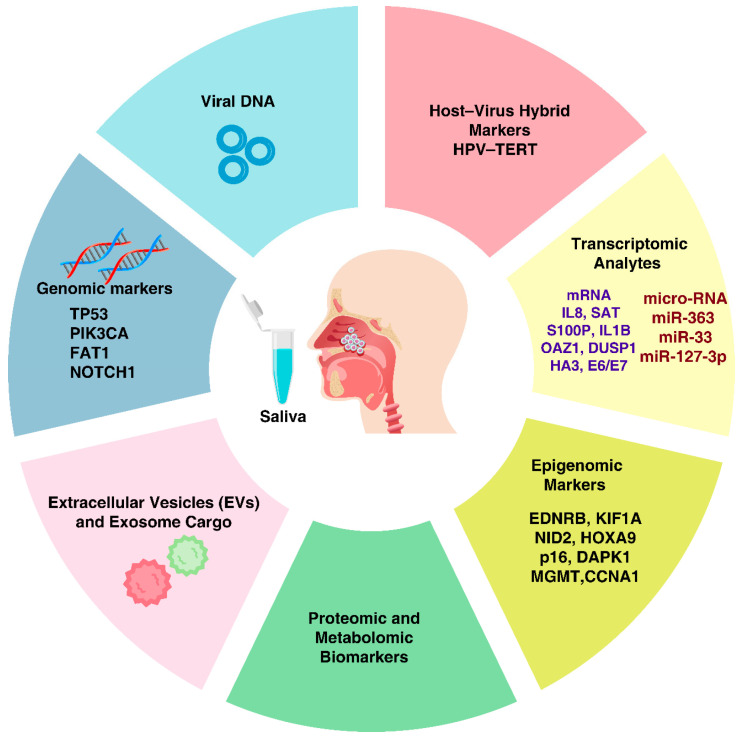
Saliva-based biomarkers for HPV-associated oral and oropharyngeal cancers. Saliva is a non-invasive diagnostic tool used to detect a wide range of biomarkers in relation to HPV-driven cancers. Biomarkers include: (i) Viral DNA, e.g., HPV DNA fragments; (ii) Genomic markers (e.g., TP53, PIK3CA, FAT1, NOTCH1); (iii) Host–virus hybrid markers, e.g., HPV–TERT; (iv) Transcriptomic analytes, e.g., mRNAs (IL8, SAT, S100P, IL1B, OAZ1, DUSP1, HA3, E6/E7) and microRNAs (miR-363, miR-33, miR-127-3p); (v) Epigenomic markers, e.g., methylated genes (EDNRB, KIF1A, NID2, HOXA9, p16, DAPK1, MGMT, CCNA1); (vi) Proteomic and metabolomic biomarkers; and (vii) Extracellular vesicles (EVs) and exosome cargo. Collectively, such molecular signatures map a complete landscape of early detection, disease monitoring, and therapeutic stratification of HPV-related oral cancers.

**Figure 3 jcm-14-07598-f003:**
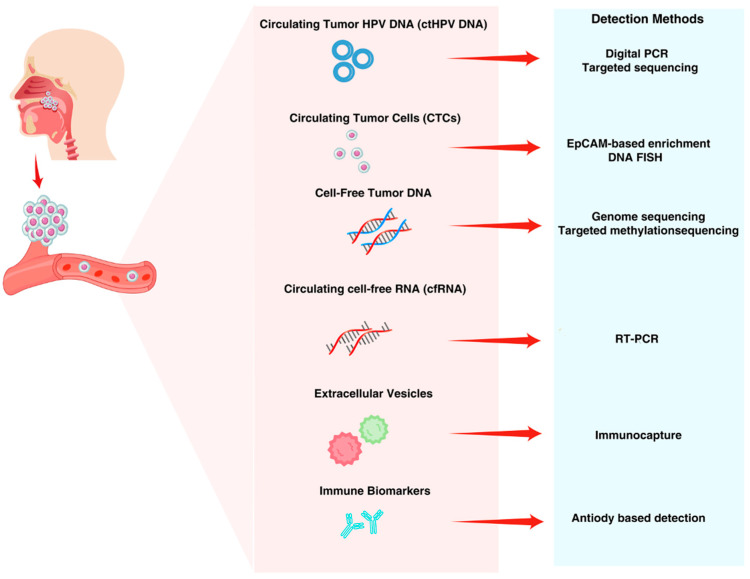
Circulating analytes are tumor-derived HPV DNA (ctHPV DNA), circulating tumor cells (CTCs), cell-free tumor DNA, circulating cell-free RNA (cfRNA), extracellular vesicles (EVs), and immune biomarkers. Different analytical methods are used for their detection: digital PCR and targeted sequencing for ctHPV DNA; EpCAM-based enrichment and DNA FISH for CTCs; genome sequencing and targeted methylation sequencing for cell-free DNA; RT-PCR for cfRNA; immunocapture for EVs; and antibody-based assays for immune biomarkers. These liquid biopsy techniques allow for minimally invasive cancer detection, monitoring, and response assessment.

**Table 1 jcm-14-07598-t001:** Cross-Tumor Concordance of Liquid Biopsy Biomarkers in Oral Cancer vs. Other Solid Tumors.

Biomarker Class	HPV+ Oral/OPSCC	Esophageal SCC (ESCC)	Non-Small Cell Lung Cancer	Breast Cancer
**Tumor Virus DNA**	HPV16/18 ctDNA is highly specific; detected in saliva & plasma of OPSCC. Enables early recurrence detection.	No analogous ubiquitous virus in ESCC (rare HPV or EBV in subsets; not routine).	No viral driver DNA in common (except rare EBV in some lymphoepitheliomas). Viral ctDNA not applicable.	No viral DNA marker (breast has no viral etiology).
**Circulating tumor DNA (mutations)**	Frequently detectable; lower mutation burden but PIK3CA, etc., appear in ctDNA. Saliva ctDNA often positive in oral cavity SCC.	Detected in plasma if advanced; common TP53 mutations can be probed. Research use (not yet clinical standard).	Well-established for genotyping (e.g., EGFR mutations for TKIs). Now used for MRD post-surgery in trials.	Under study for MRD & recurrence (e.g., PIK3CA mutations for monitoring). Not routine yet, but emerging (especially in advanced cases).
**Circulating Tumor Cells (CTCs)**	Present but rare. p16+ and HPV E6/E7 mRNA+ CTCs found in some OPSCC. No clinical use yet.	Present in advanced ESCC; EpCAM+ CTCs correlate with stage. Still research phase for clinical utility.	FDA-approved CellSearch for CTC count prognostic in metastatic (EpCAM+ CTCs)—not widely used for early lung.	FDA-approved CellSearch CTC count as prognostic in metastatic breast (≥5 CTCs = worse survival). Research on using CTCs to guide therapy.
**Extracellular Vesicles (EVs)**	High interest; salivary exosomes carry HPV DNA/RNA. EV PD-L1 and protein cargo being studied for recurrence risk.	EV biomarkers (proteins, microRNAs) explored in ESCC research, e.g., exosomal long RNAs for early detection (experimental).	Intensive research: exosomal EGFR or KRAS mutations in plasma correlating with response. Not yet in routine clinical use, but potential companion diagnostics (e.g., exo PD-L1).	Exosomal microRNA and protein signatures studied for early detection and metastasis (e.g., Her2+ exosomes). Not in clinical use yet.
**MicroRNAs (circulating)**	Saliva miR-9, miR-127, miR-363 upregulated in HPV+ tumors; panels distinguish cancer vs. control in research. Plasma miRs (e.g., miR-21) also studied.	Plasma miR-21, miR-223, others identified for ESCC diagnosis in studies—not standard.	Several plasma miRs (miR-21, miR-210, etc.) proposed as lung cancer markers; none yet guideline-approved.	miR-21, miR-155 etc. linked to breast cancer prognosis, but not used clinically; ongoing effort for miR signatures (e.g., for recurrence risk).
**DNA Methylation Markers**	Saliva: hypermethylation of tumor suppressors (p16, DAPK1, etc.) predicts oral cancer risk. Plasma: potential use in ctDNA (e.g., SOX17, ZNF genes)—research ongoing.	Several tissue markers (e.g., ZNF582) methylated in ESCC; plasma assays under exploration (early detection research).	Some blood tests in development (e.g., Lung EpiCheck) use methylation for lung CA detection; one FDA-approved lung screening blood test (2016) had methylated DNA markers.	Methylation-based multi-cancer detection (e.g., Galleri test) can pick up some breast cancers; breast-specific assays (like RASSF1A in blood) studied for recurrence.
**Unique Considerations**	Virus-driven: unique viral biomarkers enable high specificity. Local shedding: saliva tests critical. Lower overall mutation burden.	Tobacco/alcohol-driven: no virus; relies on mutation markers. Field cancerization means multiple areas may shed DNA.	Heterogeneity: high mutation load yields multiple ctDNA targets. Blood is primary fluid (no “lung saliva” equivalent). Many targeted therapies make ctDNA vital for resistance monitoring.	Multiple subtypes: (HR+ vs. Her2 vs. TNBC)—different shed patterns. CTCs more established in breast than other solid tumors.

**Table 2 jcm-14-07598-t002:** Sample Quality Metrics—Saliva vs. Blood for Liquid Biopsy.

Metric	Saliva (Oral Rinse/Swab)	Blood (Plasma)
**Invasiveness & Ease**	Non-invasive, painless collection (spit or swish). Can be self-collected in many cases. Good for frequent sampling and community screening.	Minimally invasive venipuncture. Requires phlebotomist or clinic visit. Repeat sampling limited by patient comfort and vein access.
**Typical Volume**	5–10 mL oral rinse or <2 mL saliva yields adequate DNA for assays. Volume can vary (xerostomia patients produce less).	10 mL whole blood (yields ~4–5 mL plasma) often used for ctDNA assays. Special tubes allow up to 20–30 mL draws for higher yields.
**Tumor Analyte Concentration**	Often enriched for local tumor DNA if lesion present (saliva can contain >100 pg/mL tumor DNA in oral CA). HPV DNA copies high in oral fluids for OPSCC. But background human DNA from oral mucosa and microbiome also high, diluting signal.	Generally lower absolute tumor DNA concentrations (especially in early disease). In advanced cancer, ctDNA might be 0.1–1% of total cfDNA. High background of wild-type cfDNA from normal cell turnover.
**Major Contaminants**	Food debris, oral bacteria (bacterial DNA can comprise >90% of DNA in raw saliva). Enzymes like DNases/RNases from saliva and bacteria can degrade nucleic acids if not stabilized. Viscous mucins can inhibit PCR if not purified.	Genomic DNA from leukocyte lysis (if blood not processed quickly or if using improper tubes). Hemolysis can interfere with assays (e.g., spectrophotometry) and release proteases. Anti-coagulants (EDTA or Streck tube reagents) must be appropriate for downstream assay (heparin can inhibit PCR).
**Stability (Time & Temp)**	Fresh saliva should be processed or stabilized within ~2 h for RNA; DNA is somewhat more stable but bacterial growth can alter sample. Commercial kits (Oragene, etc.) provide buffers that preserve DNA for days at ambient temp. Freeze/thaw of saliva can shear DNA, so usually one freeze recommended.	Plasma cfDNA is stable for ~6–8 h at room temp in EDTA; after that, WBC lysis dramatically increases background DNA. Specialized cfDNA tubes (Streck) stabilize samples up to ~48 h. Plasma should be double-spun and stored at –80 °C for long-term. cfDNA tolerates one freeze–thaw; CTCs/EVs require gentler handling.
**Processing Requirements**	Simple: patient can swish with buffer and spit into tube. Needs centrifugation to pellet debris if high clarity needed. Filtration can remove cells, but small tumor fragments might be lost. Often directly used in DNA extraction kits (silica columns, etc.). PCR inhibitors (mucus) might require dilution or additive (e.g., DTT).	Requires centrifugation (2-step: 1st to separate plasma, 2nd high-speed to remove any cell debris). Plasma then undergoes DNA extraction (column or beads). Volume scaling is linear—processing > 10 mL needs larger kits or multiple preps. Many labs use duplicate extractions to maximize yield.
**Analytical Challenges**	Abundance of background DNA/RNA from saliva microbiome can outcompete or confuse NGS analysis (need human-specific or viral-specific primers). High viscosity can cause sample loss in pipetting. Quantification of low-level mutant DNA tricky due to oral flora genetic diversity (false positives in shotgun sequencing).	Extremely low mutant allele fractions demand ultra-deep sequencing or digital PCR. Risk of false positives from clonal hematopoiesis (CHIP)—e.g., a TP53 mutation in plasma might originate from blood cells, not tumor. Requires matched normal controls for somatic mutation calling in NGS. PCR inhibitors minimal in plasma, but volume of data can be large (human genome background).
**Biosafety & Handling**	Generally non-infectious unless patient has oral infections. However, salivary viruses (HBV, etc.) could be present; standard precautions (PPE) recommended. Easy transport for mail-in kits (saliva tubes often treated as non-biohazard if no additive reagent).	Treat as biohazard (bloodborne pathogens). Requires phlebotomy and proper sharps disposal. Shipping regulations apply if sending plasma/serum (usually UN3373 Biological Substance Category B). Stabilizing tubes have fixatives (to prevent cell lysis)—handle chemical contents per safety sheets.
**Cost Considerations**	Very low collection cost (no personnel needed if self-collected; <$5 for a collection kit). Extraction costs similar to plasma (~$50–$100/sample for kit reagents). Overall cheaper for screening large populations.	Collection cost involves staff and phlebotomy supplies; minor (~$20). Lab processing and extraction similar cost to saliva. Often requires centralized lab for analysis (if using NGS). High-throughput processing (e.g., automated cfDNA extractors) are available but expensive initially.

## Data Availability

Data is contained within the article.

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
