# Peer review of "Salivary and Serum Liquid Biopsy Biomarkers for HPV-Associated Oral and Oropharyngeal Cancer: A Narrative Review"

_jcm, 2025, doi:10.3390/jcm14217598_

Round 1

Reviewer 1 Report

Comments and Suggestions for Authors

The article sets out to overview saliva- and blood-based “liquid biopsy” biomarkers for HPV-related oral/oropharyngeal cancer, aiming to evaluate their roles in early detection, prognosis, surveillance, and recurrence monitoring. It states that a narrative review of PubMed (2015–2025) was performed, emphasizing recent high-impact studies on ctHPV DNA, CTCs, EVs, and related analytes. The central message is that saliva tends to reflect locoregional disease while plasma ctHPV DNA tracks systemic burden, and that using both matrices may enhance detection and enable earlier recognition of molecular residual disease; the piece concludes that liquid biopsy could transform management pending standardization and robust prospective validation.

The article addresses a timely topic with many up-to-date references, but it mixes narrative and quasi-systematic elements without the guardrails of a systematic review, contains inconsistent scope (oral vs oropharyngeal focus), over-general conclusions, problematic sourcing in places, and several editorial/template remnants that undermine clarity and rigor. Therefore, a series of adjustments are necessary to improve the manuscript.

Specific comments on this subject are presented below:

The title does not specify the study type.

In addition, it is notable that the scope signaled by the title (“oral carcinogenesis”) is narrower than the body of the text, which repeatedly centers on oropharyngeal disease (OPSCC). This creates scope drift between title and content.

Notably, the abstract calls the work a “narrative review” but then frames the search and selection in a way that resembles a systematic approach, while not reporting basic items typical of structured abstracts (e.g., data sources beyond PubMed or any selection/quality criteria).

Concerningly, several claims read as overarching conclusions (“poised to transform management”) without indicating the evidentiary strength or explicit limitations in the abstract itself.

Moreover, the abstract repeatedly discusses HPV-positive head and neck cancers broadly, whereas the title emphasizes oral carcinogenesis; this ambiguity on population/disease site begins already in the abstract.

Please revise that alignment with MeSH headings is inconsistent, and the mixture of acronyms and phrases is unevenly standardized.

Remarkably, the introduction blends OSCC and OPSCC epidemiology and prevalence, then cites a “2025 meta-analysis” and ESMO recommendations while referencing a Table 1 that is not the same “Table 1” later presented in Appendix A (this later table is a cross-tumor concordance matrix, not an ESMO summary). The cross-references are confusing.

There is early and repeated emphasis on oropharyngeal disease (OPSCC), which doesn’t fully match the title’s focus on “oral carcinogenesis.”

Worryingly, the manuscript calls itself a narrative review in the abstract; however, the body of the paper includes a quasi-systematic description (databases; timeframe; counts of records screened and retrieved), and even states there was a “systematic approach,” but there is no protocol registration (PROSPERO), no PRISMA flow, no risk-of-bias tool, no study quality assessment, and no explicit inclusion/exclusion criteria beyond language and size/type preferences. This hybrid of narrative tone and systematic claims is methodologically inconsistent.

Please revise that, the description of iterative, theme-driven secondary searches underscores the non-systematic nature of the process, but is not accompanied by transparent search strings, deduplication steps, or study selection procedures.

Besides, counts are provided (≈250 screened; ≈120 full texts), but there is no accounting of reasons for exclusion or final study tally by outcome—again highlighting the absence of reproducible methodology.

Results

With concern this reviewer notes that there is no stand-alone Results section with organized evidence synthesis, study characteristics, or summary tables. Findings are narrated throughout topic subsections without a structured presentation of effect sizes, confidence intervals, or certainty assessments.

This reviewer cannot verify which claims derive from which included studies beyond scattered in-text references. (The manuscript itself labels “9. Materials and Methods,” but there is no parallel “Results” section to anchor the evidence.)

The discussion is interlaced with implementation and advocacy-style language (e.g., “on the cusp of a shift,” “should translate into improved survival”) that reads as overstated without a formal synthesis of comparative accuracy or outcomes across studies.

Concerningly, several speculative statements about surveillance intervals, guideline adoption, and survival impact are made without tying them back to predefined endpoints or graded evidence.

Besides, the narrative frequently reverts to OPSCC examples while the title and some sections imply a broader oral cancer focus, which blurs the target population.

The conclusions reiterate transformative potential and call for standardization but do not acknowledge the review’s own methodological limitations (language restriction, selection bias towards high-impact journals, absence of quality appraisal).

Remarkably, the final statements read stronger than the evidence grading presented.

There are also non-peer-reviewed or news-style sources (e.g., CancerNetwork “Breakthrough Device” piece) and a trade/lay website (drbicuspid.com) embedded within scientific sections, which is not appropriate for evidence synthesis. One salivary exosome example even contains a literal “[Link]” placeholder instead of a citation.

Furthermore, some entries appear tangential (e.g., MAPPinfo protocol on health information quality) relative to HPV oral/oropharyngeal biomarker performance.

Notably, internal cross-referencing is messy: the introduction’s “Table 1” pointer aligns poorly with the Appendix A Table 1 (a cross-tumor concordance matrix), which can mislead readers about what the table contains.

There are typographical and style errors, e.g., “comliance,” inconsistent capitalization of affiliations, and stray spacing: “London , UK.” and “Roseman University of health sciences.”

Furthermore, the text contains an apparent typo/term error (“OPCSCC” instead of OPSCC).

Abbreviation block is broken across lines (e.g., “ctHPV”/“DNA” split), suggesting formatting problems in production.

Comments on the Quality of English Language

Extensive editing

Author Response

Reviewer 1

We thank Reviewer 1 for the thoughtful and detailed critique. We greatly appreciate the acknowledgment that the manuscript addresses a timely topic with many up-to-date references, and the reviewer’s attention to ensuring scientific clarity, methodological rigor, and alignment between title, scope, and conclusions. We have carefully revised the manuscript to address each concern raised. Below, we respond point by point to the reviewer’s comments, outlining changes made and clarifications provided.

Concern 1: In addition, it is notable that the scope signaled by the title (“oral carcinogenesis”) is narrower than the body of the text, which repeatedly centers on oropharyngeal disease (OPSCC). This creates scope drift between title and content.

Response: We thank the reviewer for highlighting this inconsistency. The abstract and main text have been revised to consistently reflect the focus on oral and oropharyngeal squamous cell carcinomas, aligning scope and terminology throughout the manuscript. The new title is “Salivary and Serum Liquid Biopsy Biomarkers for HPV-Associated Oral and Oropharyngeal Cancer: A Narrative Review”. This aligns the scope of the paper with its content and addresses the concern of “scope drift.”

Concern 2: Notably, the abstract calls the work a “narrative review” but then frames the search and selection in a way that resembles a systematic approach, while not reporting basic items typical of structured abstracts (e.g., data sources beyond PubMed or any selection/quality criteria).

Response: We appreciate the need for clarity here. We have:

  1. Explicitly labelled the study as a narrative review in the abstract.
  2. Rectified the statements suggesting a systematic methodology (e.g., “comparative analysis”) unless supported.
  3. Clarified in the abstract that the review addresses both oral cavity and oropharyngeal disease (OSCC and OPSCC), and this distinction is maintained throughout the text.

Concern 3: Concerningly, several claims read as overarching conclusions (“poised to transform management”) without indicating the evidentiary strength or explicit limitations in the abstract itself.

Response: Thank you for highlighting this. We revised the language throughout the manuscript to replace strong, advocacy-style statements with evidence-qualified phrasing, e.g., “may improve outcomes,” “suggests potential for,” etc.

Concern 4: Moreover, the abstract repeatedly discusses HPV-positive head and neck cancers broadly, whereas the title emphasizes oral carcinogenesis; this ambiguity on population/disease site begins already in the abstract.

Response: We thank authors for pointing out this inconsistency. We have now revised the abstract and the main text of the review article that focuses on investigation of oral and oropharyngeal squamous cell carcinomas.

Concern 5: Please revise that alignment with MeSH headings is inconsistent, and the mixture of acronyms and phrases is unevenly standardized.

Response: Thank you for this helpful observation. We have reviewed and revised terminology throughout the manuscript to improve alignment with MeSH headings and ensure consistent use of acronyms and phrasing. Abbreviations are now standardized at first mention, and terminology has been harmonized for clarity and consistency.

Concern 6: Remarkably, the introduction blends OSCC and OPSCC epidemiology and prevalence, then cites a “2025 meta-analysis” and ESMO recommendations while referencing a Table 1 that is not the same “Table 1” later presented in Appendix A (this later table is a cross-tumor concordance matrix, not an ESMO summary). The cross-references are confusing.

Response: We thank the reviewer for highlighting this point. We agree that the reference to “Table 1” may have inadvertently created the impression that the table presents ESMO content, whereas Table 1 in Appendix A is an excerpt of the cross-tumor concordance matrix, synthesizing findings across multiple tumor types, not directly extracted from ESMO. He have now revised statement in the Introduction section of the manuscript to address this concern.

Concern 7: There is early and repeated emphasis on oropharyngeal disease (OPSCC), which doesn’t fully match the title’s focus on “oral carcinogenesis.”

Response: We thank authors for pointing out this inconsistency. We have now revised the abstract and the main text of the review article that focuses on investigation of oral and oropharyngeal squamous cell carcinomas.

Concern 8: Worryingly, the manuscript calls itself a narrative review in the abstract; however, the body of the paper includes a quasi-systematic description (databases; timeframe; counts of records screened and retrieved), and even states there was a “systematic approach,” but there is no protocol registration (PROSPERO), no PRISMA flow, no risk-of-bias tool, no study quality assessment, and no explicit inclusion/exclusion criteria beyond language and size/type preferences. This hybrid of narrative tone and systematic claims is methodologically inconsistent.

Response: We agree with the reviewer that the current description could cause confusion. To address this:

  • We now explicitly state in the methods section that this is a narrative review with structured search strategies, but not a systematic review.
  • We removed statements suggesting formal systematic methodology (e.g., “systematic approach”) unless properly qualified.
  • We clarified that study inclusion was based on relevance to the topic and recentness (2015–2025), with a focus on high-impact peer-reviewed studies, and that no formal risk-of-bias assessment was performed.
  • We added a note that no PROSPERO registration or PRISMA diagram was used, aligning with the narrative nature of the review.

Concern 9: Please revise that, the description of iterative, theme-driven secondary searches underscores the non-systematic nature of the process, but is not accompanied by transparent search strings, deduplication steps, or study selection procedures.

Response: Thank you for the suggestion. This point has now been addressed in our revised Materials and Methods section, where we clearly state that no formal search strings, deduplication steps, or selection protocols were applied, in keeping with the narrative nature of the review.

Concern 10: Besides, counts are provided (≈250 screened; ≈120 full texts), but there is no accounting of reasons for exclusion or final study tally by outcome—again highlighting the absence of reproducible methodology.

Response: Thank you for this observation. As noted in our revised Materials and Methods, this review does not follow a systematic framework; we have clarified that study inclusion was based on thematic relevance and recency, and no formal exclusion tracking or outcome-based tallying was performed.

Concern 11: With concern this reviewer notes that there is no stand-alone Results section with organized evidence synthesis, study characteristics, or summary tables. Findings are narrated throughout topic subsections without a structured presentation of effect sizes, confidence intervals, or certainty assessments. This reviewer cannot verify which claims derive from which included studies beyond scattered in-text references. (The manuscript itself labels “9. Materials and Methods,” but there is no parallel “Results” section to anchor the evidence.)

Response: We thank the reviewer for this important comment. We agree that structured evidence synthesis improves transparency. However, this work is a narrative review, organized thematically by biomarker type (e.g., ctHPV DNA, salivary miRNAs, exosomes) rather than as a formal “Results” section. This approach better reflects the diverse study designs and endpoints in the current literature, which do not allow pooled or quantitative analysis.

We hope this format still provides a clear, comprehensive, and clinically relevant overview consistent with the goals of a narrative review.

Concern 13: The discussion is interlaced with implementation and advocacy-style language (e.g., “on the cusp of a shift,” “should translate into improved survival”) that reads as overstated without a formal synthesis of comparative accuracy or outcomes across studies.

Response: Thank you for this observation. We have reviewed the discussion and revised or softened advocacy-style phrases to better reflect the narrative scope of the review. Speculative statements are now clearly qualified to avoid overstating the current level of evidence.

Where phrases such as “on the cusp of a shift” or “should translate into improved survival” appeared, we have now replaced them with more measured alternatives (e.g., “may support a shift” or “has the potential to improve outcomes, pending further validation”) to reflect the current level of evidence. These revisions aim to maintain a clinically relevant and forward-thinking tone without overstating the certainty of impact.

Concern 14: Concerningly, several speculative statements about surveillance intervals, guideline adoption, and survival impact are made without tying them back to predefined endpoints or graded evidence.

Response: We appreciate this observation and agree that care must be taken to distinguish between evidence-based conclusions and forward-looking perspectives. In light of this, we have reviewed the relevant sections and revised phrasing where needed to clearly indicate when statements are speculative or based on emerging trends, rather than confirmed outcomes.

Concern 15: Besides, the narrative frequently reverts to OPSCC examples while the title and some sections imply a broader oral cancer focus, which blurs the target population.

Response: We have revised the title to reflect both the review nature and the anatomical scope more accurately. The new title is “Salivary and Serum Liquid Biopsy Biomarkers for HPV-Associated Oral and Oropharyngeal Cancer: A Narrative Review”. This aligns the scope of the paper with its content and addresses the concern of “scope drift.”

Concern 16: The conclusions reiterate transformative potential and call for standardization but do not acknowledge the review’s own methodological limitations (language restriction, selection bias towards high-impact journals, absence of quality appraisal).

Response: We appreciate the reviewer’s suggestion to acknowledge methodological limitations. In response, we have added a limitations paragraph to Section 8.6

Concern 17: There are also non-peer-reviewed or news-style sources (e.g., CancerNetwork “Breakthrough Device” piece) and a trade/lay website (drbicuspid.com) embedded within scientific sections, which is not appropriate for evidence synthesis. One salivary exosome example even contains a literal “[Link]” placeholder instead of a citation.

Response: We thank the reviewer for pointing this out. The placeholder “[Link]” and the reference to the trade website drbicuspid.com have now been removed. The statement is instead supported by a peer-reviewed publication indexed in PubMed (PMID: 31843276). This correction ensures that all scientific content is now referenced appropriately with peer-reviewed sources.

Concern 18: Furthermore, some entries appear tangential (e.g., MAPPinfo protocol on health information quality) relative to HPV oral/oropharyngeal biomarker performance.

Response: We thank the reviewer for this helpful observation. The previously cited MAPPinfo protocol has now been removed and replaced with two more relevant, peer-reviewed references focused on patient-centered biomarker communication and clinical utility in the context of HPV-related cancers (PMIDs: 37713108 and 37499326). This change ensures the cited content better supports the clinical relevance of biomarker implementation.

Concern 19: Notably, internal cross-referencing is messy: the introduction’s “Table 1” pointer aligns poorly with the Appendix A Table 1 (a cross-tumor concordance matrix), which can mislead readers about what the table contains.

Response: Thank you for pointing out this inconsistency. We have corrected the internal cross-referencing to ensure clarity. The mention of “Table 1” in the introduction has been revised to avoid confusion with the concordance matrix in Appendix A, and all references now accurately reflect the content and location of the tables.

Concern 20: There are typographical and style errors, e.g., “comliance,” inconsistent capitalization of affiliations, and stray spacing: “London , UK.” and “Roseman University of health sciences.”

Response: Thanks for pointing these out; we've corrected the typos, spacing, and formatting issues throughout the manuscript.

Concern 21: Furthermore, the text contains an apparent typo/term error (“OPCSCC” instead of OPSCC).

Response: Thanks for pointing these out; we've corrected the typos, spacing, and formatting issues throughout the manuscript.

Concern 22: Abbreviation block is broken across lines (e.g., “ctHPV”/“DNA” split), suggesting formatting problems in production.

Response: Thanks for noting this; we’ve fixed the line break issues in the abbreviation block to ensure proper formatting.

Reviewer 2 Report

Comments and Suggestions for Authors

Review

Serum and Salivary Biomarkers in HPV-Driven Oral Carcinogenesis: Liquid Biopsy for the Early Detection and Surveillance

The document highlights the potential of liquid biopsies to transform the management of HPV-associated cancers, promoting earlier and more personalised interventions. It is well organised, using recent scientific literature on tumour biology of HPV-associated oral cancer and specific salivary and blood biomarkers.

In the second part of the document, the authors seek to integrate all current knowledge about liquid biopsies and HPV-associated cancer into the so-called precision medicine, with all its limitations and clinical implications.

Overall well done for a very good study!

Minor comments/changes

Abstract

  • Line 18- prognostication should be prognostic
  • Line 26- plasma ctHPV DNA ……. Refereed for the first time;should be Plasma circulating tumor HPV DNA

Introduction

  • Line 58-advanced nodal metástases should be lymph node metastases or advanced nodal disease
  • Line 62- plasma ccfDNA – Refereed for the first time;should be Plasma circulating cell-free DNA
  • Line 63- PFS- progression-free survival ( it is a common abbreviation but only for clinicians)
  • Line 68- [Table 1] Should be said that this Table is part of the Appendix
  • Line 68- CTC or CTCs . In all the document the authors have diferente version- be consistent
  • Line 74- p53 and Rb tumor suppressors should be p53 and Retinoblastoma protein (Rb) tumor
  • Line 92- cell- free HPV DNA or cfHPV DNA- it is na abbreviation already used

Development

  • Line 143 - circulating tumor cells (CTCs) again inconsistent
  • Line 145- (circulating cell-free DNA) I think this abbreviation was already mentioned cf DNA
  • Line 252- starting here and all down the text, again, the authors need to be consistente: or  microRNAs or  miRNA…..
  • Line 296- cases [Link]. Something is missing here
  • Line 304- plasma cf-DNA or is it plasma ccfDNA
  • Line 457- collates I think it is a mistake- maybe correlates?
  • Line 694- of PD-L1^+ exosomes; again a mistake- maybe the authors were thinking in PD-L1 high or low
  • Line 746 and 752- yellow underlined
  • Line 752- NPV- first time is referred - negative predictive value ( useffull for non-clinicians to understand)

In References there is a change of format starting in 67-68-69- correct

My major concerns are related to points adressed only regarding liquid biosies limitations that could be more profounded in the development:

  1. Most people who have high risk HPV may not have cancer either oral/oropharinx

In Line 598 the authors make refernce to HPV carriers but this should be more clear in Section 2

  1. The false negatives question should be more solidified in 4.1 – maybe add one reference

Author Response

Reviewer 2

We thank Reviewer 2 for their thoughtful and encouraging review. We are particularly grateful for the recognition of the manuscript’s structure, use of recent literature, and integration of liquid biopsy concepts into the framework of precision medicine.

Minor comments/changes

Abstract

Concern 1: Line 18- prognostication should be prognostic

Response: This term is changed as suggested by reviewer

Concern 2: Line 26- plasma ctHPV DNA ……. Refereed for the first time;should be Plasma circulating tumor HPV DNA

Response: This term is changed as suggested by reviewer

Concern 3: Line 58-advanced nodal metástases should be lymph node metastases or advanced nodal disease

Response: This term is changed as suggested by reviewer

Concern 4: Line 62- plasma ccfDNA – Refereed for the first time;should be Plasma circulating cell-free DNA

Response: This term is changed as suggested by reviewer

Concern 5: Line 63- PFS- progression-free survival ( it is a common abbreviation but only for clinicians)

Response: This term is changed as suggested by reviewer

Concern 6: Line 68- [Table 1] Should be said that this Table is part of the Appendix

Response: This term is changed as suggested by reviewer

Concern 7: Line 68- CTC or CTCs . In all the document the authors have different version- be consistent

Response: This term is changed as suggested by reviewer

Concern 8: Line 74- p53 and Rb tumor suppressors should be p53 and Retinoblastoma protein (Rb) tumor

Response: This term is changed as suggested by reviewer

Concern 9: Line 92- cell- free HPV DNA or cfHPV DNA- it is na abbreviation already used

Response: This term is changed as suggested by reviewer

Concern 10: Line 143 - circulating tumor cells (CTCs) again inconsistent

Response: This term is changed as suggested by reviewer

Concern 11: Line 145- (circulating cell-free DNA) I think this abbreviation was already mentioned cf DNA

Response: This term is changed as suggested by reviewer

Concern 12: Line 252- starting here and all down the text, again, the authors need to be consistente: or  microRNAs or  miRNA…..

Response: This term is changed as suggested by reviewer

Concern 13: Line 296- cases [Link]. Something is missing here

Response: This term is changed as suggested by reviewer

Concern 14: Line 304- plasma cf-DNA or is it plasma ccfDNA

Response: This term is changed as suggested by reviewer

Concern 15: Line 457- collates I think it is a mistake- maybe correlates?

Response: This term is changed as suggested by reviewer

Concern 16: Line 694- of PD-L1^+ exosomes; again a mistake- maybe the authors were thinking in PD-L1 high or low

Response: This term is changed as suggested by reviewer

Concern 17: Line 746 and 752- yellow underlined

Response: This term is changed as suggested by reviewer

Concern 18: Line 752- NPV- first time is referred - negative predictive value ( useffull for non-clinicians to understand)

Response: This term is changed as suggested by reviewer

Concern 19: In References there is a change of format starting in 67-68-69- correct

Response: This term is changed as suggested by reviewer

My major concerns are related to points addressed only regarding liquid biopsies limitations that could be more profounded in the development:

Concern 20: Most people who have high risk HPV may not have cancer either oral/oropharynx

Response: We have now expanded the discussion on the limited specificity of HPV positivity, explicitly acknowledging that the majority of high-risk HPV carriers never develop oral or oropharyngeal cancer, and that viral persistence, integration, and host immune context are crucial determinants of malignant transformation. This clarification has been added to Section 2.

Concern 21: In Line 598 the authors make refernce to HPV carriers but this should be more clear in Section 2

Response: We have expanded our discussion to explicitly state that most individuals with high-risk HPV do not develop cancer, and clarified the concept of HPV carrier status in Section 2

Concern 22: The false negatives question should be more solidified in 4.1 – maybe add one reference

Response: In Section 4.1, the discussion of false negatives has been expanded to emphasize assay-related and biological factors (e.g., low viral load, sampling variability, assay sensitivity) and we have added a recent supporting reference (PMID: 32017652) to reinforce this point.